# Using Dancesport as an Educational Resource for Improving Institutionalized Children’s Learning Strategies

**DOI:** 10.3390/children10061039

**Published:** 2023-06-09

**Authors:** Gabriela Tomescu, Monica-Iulia Stănescu, Mihaela Manos, Liliana Dina, Kamer-Ainur Aivaz

**Affiliations:** 1Faculty of Physical Education and Sport, Ovidius University of Constanța, 900470 Constanța, Romania; tomescu.gabriela03@yahoo.com; 2Faculty of Physical Education and Sport, National University of Physical Education and Sport, 060057 Bucharest, Romania; manosmihaela@yahoo.com (M.M.); lili.dina@yahoo.com (L.D.); 3Faculty of Economic Sciences, Ovidius University of Constanța, 900001 Constanța, Romania; kamer.aivaz@365.univ-ovidius.ro

**Keywords:** learning strategies, institutionalized children, multiple intelligences, dance

## Abstract

Introduction: Specialized studies mention that extracurricular activities (including dance) contribute to the stimulation of multiple intelligences, on whose development the educational process and academic success depend. The aims of the study were to investigate the benefits of dancesport for the development of institutionalized children’s learning strategies, and to examine gender-dependent differences in learning strategies, as well as to formulate possible recommendations regarding the practice of dance at the age of preadolescence, from the perspective of school success vectors. Methods: Through the School Motivation and Learning Strategies Inventory (SMALSI), we could observe the changes produced in children from the initial phase to the final assessment at the end of a dancesport program. The intervention took place over a period of six months with a frequency of two lessons per week, with each lesson lasting 60 min, and aimed to increase school motivation and performance, considering the learning strategies used by institutionalized children. Thirty institutionalized children, aged 11–12 years old, participated in the research, during which they did not engage in other extracurricular physical activities. The preadolescents were assessed using the School Motivation and Learning Strategies Inventory (SMALSI). This self-report rating scale measures nine areas associated with learning strategies, six of which focus on student strengths (study strategies, note-taking/listening skills, reading/comprehension strategies, writing/research skills, test-taking strategies, and time management/organization techniques), and three are aimed at student liabilities (low academic motivation, test anxiety, and concentration/attention difficulties). Results: The results show that the biggest improvements in the case of institutionalized children were recorded for study strategies, effectiveness of test-taking strategies, and concentration difficulties. Girls registered significantly better results than boys only in the case of study strategies and for writing/research skills (Mann–Whitney test was used). Discussion: The study demonstrates the benefits of dancesport practice for the development of institutionalized children’s learning strategies, creating a foundation for the improvement of their academic performance and school integration. Conclusions: At the end of the dance program, significant improvements in academic interest were observed due to the testing strategies used. Better results were also obtained for the scales of anxiety and difficulty concentrating during tests, where average scores decreased significantly.

## 1. Introduction

Experts claim that physical activity is essential for children’s development, and movement integration into the educational system should be a priority because the current pedagogical methodology involves sedentary activities in a proportion of approximately 89%. The academic content of a curriculum that is supported by physical exercise improves school performance, which is why specialists recommend that the educational system should include physical activity for at least one and a half hours per day [1].

However, usually, the reduced number of lessons does not allow for taking advantage of all the influences that derive from the practice of exercise. For this reason, leisure physical activity plays an important role in the process of growth and development, and school and family have a decisive impact on promoting a healthy lifestyle for children [2]. 

In a physical and physiological sense, childhood marks the most important developmental stage, during which personality is formed [3]. The influence of the growth environment and the activities carried out are crucial for the physical and mental development of children, and emotional control plays a key role in the formation of a healthy lifestyle in cognitive (subjective), behavioral (expressive), and psychological (adaptive) terms [4], which is why it should be stimulated both during school hours and in leisure time. 

Motivation and school success depend on certain internal dynamic factors, such as skills and motives to satisfy the need for success [5]; however, the training process decisively benefits from motor activities and the support provided by educators in the direction of perception, orientation, and awareness [6].

Through their specific content, sports activities influence the path to general education in motor, intellectual, esthetic, and emotional terms [7], contributing to the formation of interests and motivation for obtaining good results in other disciplines and fields, but also to the bio-psychological development of the child [8].

Compared to people who practice other physical and leisure activities, dancers and musicians have a greater ability to distinguish sounds, understand information, anticipate and imitate the next movements of people around them or other living organisms, feel the rhythm and synchronize movements to music, orient themselves in space and time, and control their posture [9]. Additionally, they are more concerned with professional recognition and relationships with those around them [10].

Dance is a physical activity that includes various forms of expression. Its complexity and aerobic nature provide health benefits, reducing excess weight (which increases the risk of chronic illnesses such as cancer, diabetes, and cardiovascular disease), correcting posture, and strengthening muscles [11].

According to the literature, dancesport is, due to the countless functions it fulfills, at the border between sport and art, being a motor activity expressed nonverbally both in the sphere of bodily expression and from a narrative point of view by performing rhythmic movements based on choreographic scenarios that involve multiple movements of the body such as travels, twists, leaps, turns, flexion and extension, etc. [12].

The variety of functions and benefits offered by sports dance organizes it into three distinct categories: elite dance, dance for all (whose purpose is to achieve an optimal state of health and harmonious physical development, educate body esthetics, and develop conditional and coordination skills), and adapted dance (the compensatory effects of dance contribute to the correction of defects) [13].

Starting from these ideas, the question arises as to how the educational value of leisure time can be increased for institutionalized children, considering that in Romania they still represent a vulnerable population category at high risk of educational and social exclusion [14]. Although national programs for the integration of children from disadvantaged backgrounds have been implemented, the lack of coherent measures and the inconsistency of resources do not allow significant outcomes to be obtained in this regard. The effect of an unmotivating educational system is reflected in the number of students who lose interest in school [15]. One way to make sure that every child gets the best lessons for their well-being is to make them part of both the regular school day and their home life. Goleman and Lantieri are co-founders of the Collaborative for Academic, Social and Emotional Learning, an organization established at the University of Illinois in Chicago, which has set standards for SEL (social and emotional learning) and helped school systems from around the world to introduce these programs to their curricula. The best SEL programs are designed to fit perfectly into the standard school curriculum of children of all ages. This project provides a final meta-analysis of more than one hundred studies that compared students who received SEL with those who did not. Data show impressive improvements in behavior in and out of the classroom for children who have benefited from SEL. Students not only enhanced their relaxation and socializing skills, but also learned more effectively; their grades improved and their academic achievement test scores were a significant 14 percent higher than those of students in the same year who had not received such a social and emotional learning program. By helping children manage their emotions and relationships, they are helped to learn better [16].

That is why extracurricular offerings are even more important for institutionalized children and require an increased involvement in education or protection institutions’ training. Therefore, solutions are being sought to complement the compulsory core curriculum with a number of extracurricular activities that would enhance the educational influence of the formal environment.

Education specialists have tested different strategies and proposed varied content. The inclusion of dance in the educational experience of Australian students is becoming more widespread. The new national curriculum incorporates dance, drama, media arts, music, and visual arts as compulsory arts-focused learning areas that are essential to students’ first eight years of education [17]. Certain intervention programs or activities can protect institutionalized children from negative consequences and contribute to their harmonious physical development [18]. However, no studies have been conducted so far in Romania to verify the effectiveness of dance practice for the development of institutionalized children. In order to create arguments for this content and to influence the development of these children, the present study aims to highlight the role of dance in the formation of their learning strategies and the potential of this leisure activity to contribute to a better educational integration of institutionalized children.

### 1.1. The Effects of Institutionalization on Children’s Development

This study is very important for the educational approach to institutionalized children because it addresses an important school problem—low academic motivation and, as a possible consequence, drop-out, which is influenced, also, by pre-teens’ value structures. There is a connection between personality type and personal interests, and the relationships between them are reflected in children’s school situations [19].

Parental behavior towards children and the environment in which they develop greatly influence their mental health, which contributes to their formation as adults. Thus, social behavior and educational and professional success depend on the education received in childhood [3].

Parental support is an external factor of influence without which school failure and poor cognitive development can occur: children are not assisted in doing their homework and may become unable to study further. Social and school environments can also have an impact on students’ school motivation [20]. Dysfunctional family life can make children vulnerable when faced with the norms of institutionalized education after the age of 6–7 years old. Previous attachment disorders can also manifest as school phobia or teacher phobia. School failure affects individual cognitive strategies and self-control ability [21]. In institutionalized children, frustration and negative thoughts are more pronounced, and school performance is less important to them [22].

The negative effects of adverse childhood experiences are also reflected in each person’s mental health in different proportions. A study of 12,421 adolescents aged 10–17 suggests that girls are more affected by negative childhood experiences compared to boys, but boys experience more violence. These children are more likely to develop depression and anxiety symptoms, especially girls [23].

### 1.2. Dance as an Educational Resource 

The importance of dance in education has been highlighted for many years, with numerous authors demonstrating the need to include it in the school curriculum. In Sarasota, Florida, dance programs have been developed for high school students, especially those from disadvantaged backgrounds, with the purpose of encouraging them through this form of movement to complete their studies. Strategies were applied to improve the education process through art and music, and after about two years, it was observed that the results were much improved in terms of school performance [24].

The influence of dance on the learning process is also mentioned in studies emphasizing that dance training involves pedagogical teaching methods and that a dancer needs to have minimal notions of anatomy. Such an activity improves mental health and facilitates the development of other curricular and extracurricular activities in good conditions of concentration and attention [25].

Although society believes that artistic sports would be more suitable for girls than boys, specialists in the field claim that these activities should be equally addressed to both genders as an everyday activity [26]. The same study recommends the practice of dance by institutionalized children because it guides towards a specific career, adapts the body to usual and unusual daily demands, stimulates nonverbal communication, develops spatial orientation, and improves the social integration process.

Dance stimulates non-cognitive abilities; therefore, artistic and social skills are better developed in people who practice this sport compared to those who do not perform any physical activity in their free time [27,28]. Recent studies have demonstrated the effectiveness of including dance in the school curriculum due to its complex effects on children’s growth and development as well as its psychological benefits [29]. Dance can be practiced with a therapeutic purpose: it improves the health status of people with various mental or motor conditions, increases school or professional results, and prevents the deterioration of brain activity [30]. The enjoyment of music and dance plays an important role in increasing the individual’s daily motivation [31].

In this context, physical education programs should be better exploited by educational institutions, and emotional and social factors should represent benchmarks for further research and intervention studies [32].

This approach is particularly important for successful education because multiple intelligences influence the learning process and career orientation. A group of eleventh-grade students participated in a study addressing their ability to learn biology and the connection of this process with multiple intelligences [33]. Kinesthetic intelligence seems to influence learning ability in this subject, with the information about anatomy that students receive in school being easier to remember for people who are characterized by this type of intelligence. 

Other studies have highlighted the contribution of dance to developing kinesthetic intelligence and therefore improving postural, sensorimotor, and cognitive performance. A sample of 31 experienced dancers aged 11–54 took part in a six-month study that involved three training sessions per week, with each class lasting 90 min. At the end of this program, improvements were found in body control and reaction time [34]. 

The emotional intelligence of professional dancers is very well developed, especially in those practicing dance from a young age, who have learned early in life to strategically use their emotions in various contexts, allowing them to be persistent, self-confident, efficient, precise and methodical, focused, energetic, patient, tolerant, sociable, empathetic, and fair [35]. Stimulating this type of intelligence through dance since childhood also involves the formation of multiple interests in other activities and the establishment of professional objectives [36].

Body expression is defined as a phenomenon consisting of cognitive processes, motivations, attitudes, mental states, and various personality traits that contribute to the psychomotor development of children, given that dance stimulates body expressivity and musicality [37]. Therefore, including this sport in the school curriculum is beneficial for children’s personality development [38]. A recent study [39] confirms the importance of dance (as art therapy content) for children’s mental health and emotional well-being. A group of 62 primary school children with mild emotional and behavioral difficulties were randomly assigned to art therapy, music therapy, dance movement therapy, or dramatherapy and, at the end of the experiment, positive changes were noted in their self-esteem, sense of safety, and optimism for the future, as well as artistic and verbal progress.

The theoretical background for these explanations is based on Gardner’s Theory of Multiple Intelligences [40]. According to this theory, there are several intelligences that manifest simultaneously at different levels; they are used concurrently and complement each other as individuals develop their cognitive and non-cognitive skills. Every person has multiple intelligences that can be developed through education and various activities [41]. The identified intelligences are: verbal/linguistic, logical/mathematical, visual/spatial, bodily/kinesthetic, musical/rhythmic, interpersonal/social, intrapersonal/emotional, and naturalistic.

Everyone has abilities in all types of intelligence, but they exist in different proportions and function in unique ways for each individual. There are people who show a high level of functionality in all types of intelligence, or it may be that only some of them have maximum results, while the results of others are moderate or minimal. Unfortunately, the educational system places emphasis on verbal and mathematical skills [42]. Compared to the traditional school teaching model, the model based on the development of multiple intelligences requires the teacher to combine intelligences in a creative way to stimulate linguistic, musical, spatial, logical, and other types of intelligence [42]. 

Studies in the field have described the importance of dance in the development of various types of intelligence as follows:Verbal Intelligence: verbal communication is necessary in this sport to express everyone’s wishes regarding the choreography to be performed. Collaboration should be effective both between partners and between dancers and coaches [43].Logical Intelligence: the logical chaining of steps, accompanied by technical and artistic elements, is a means of developing mathematical intelligence [44].Bodily Intelligence: dance improves postural, sensorimotor, and cognitive performance [34].Musical Intelligence: dance is based on an attitude of listening to all the surrounding sounds that, following personal perceptions, are expressed physically and emotionally to a musical rhythm [45].Social Intelligence: dance involves respect and understanding towards other people, and their acceptance and collaboration with the partner or the group to which the individual belongs [46].Emotional Intelligence: people educated through music and rhythmic activities have a deeper understanding of the relationships between feelings and the environment, which stimulate the ability to express personal feelings, create, and listen [47].Naturalistic Intelligence: dance develops naturalistic intelligence by carefully listening to the sounds of nature (flora and fauna) and imitating them through this activity [48].Aesthetic Intelligence: this type of intelligence suggests the artistic sensibility that a person can bring to their creation and work. Any dancer should always be authentic and original [49].In terms of visual/spatial intelligence, the individual acquires the ability to think in images, to either clearly visualize the examples provided by the instructor or abstractly visualize them based on imagination, which will be adapted to the workspace or the movement possibilities [50].

The earlier dance education begins, the greater the chances of developing intelligence. Brain activity is extremely important in children, and this sport has a positive mental, emotional, and social impact from the first years of life, with the forms of communication in that period relying on expression through gestures and movements [51].

Previous studies demonstrated the effectiveness of dance for the harmonious development of preadolescents from a physical, mental, intellectual, social, and emotional point of view. Based on the Theory of Multiple Intelligences, the dancesport program for the institutionalized children participating in this study is designed so as to be adapted to all types of intelligence theorized so far. Given that dance is a complex sport that includes a variety of styles from the Latin-American and European categories, it has the quality of simultaneously improving several aspects of development, which contributes to the overall development of an individual.

The School Motivation and Learning Strategies Inventory used in this research highlights the changes in institutionalized children in terms of school success, contributing to the assessment of certain types of intelligence, especially linguistic, logical, and emotional intelligences.

### 1.3. Multiple Intelligences and Learning Strategies

Including the model of multiple intelligences in the education of children can have a positive effect on their learning strategies, increasing their motivation and academic interest [52]. 

There are schools that have already applied the Theory of Multiple Intelligences in the teaching process and reported significantly improved performance in academic achievement tests; thus, students’ scores increased by 20% in a Maryland school after just one year of implementing the new educational model. According to a study conducted with 288 fourth-grade children, other schools in Chicago also found a correlation between logical/mathematical intelligence and reading comprehension skills [53]. 

To improve school results, the learning process can be stimulated by exploiting multiple intelligences [54], as shown in Table 1.

All these variables (Table 1) contribute to achieving academic success in accordance with learning strategies, the way of doing homework, and the efficient organization of the time allocated to study and extracurricular activities [55]. The same study mentions that emotional stability plays a very important role in terms of concentration and attention to schoolwork, while anxiety and negative emotions are associated with absenteeism and low academic performance. Extroverts have better school results due to their increased energy and positive attitudes, these characteristics being accompanied by a desire to understand the lessons and assimilate the information taught by the teacher. 

There are children who fail to obtain passing grades in some exams because of just one section, although they have learned and could verbalize the responses to items. However, they are not able to focus enough during tests or do not understand the requirements of certain items. This is due to the types of dominant intelligences, which is why specialists consider it necessary to evaluate children differently so as to highlight students’ qualities and thus encourage them to use their skills outside the school too [56]. A test should not simply involve writing on paper the acquired knowledge, it can also be expressed in the form of video projects that include linguistic, musical, and spatial elements (pictures, graphs). If children were evaluated according to their skills, the average number of students with school achievements would increase as a result of improved learning strategies [41]. 

In a 2015 study conducted in a Canadian school [57], the School Motivation and Learning Strategies Inventory was administered to a group of 404 sixth-grade students. The authors specify that writing and research skills are the most significant for the accumulation of information from a variety of sources and the educational development of students. Achieving satisfactory outcomes depends on skill development, effective time management and stress management. A group of 1,300 students from three schools in Florida participated in a study aimed at assessing their learning strategies, after which they reported that the School Motivation and Learning Strategies Inventory (SMALSI) had helped them realize the importance of setting goals and thinking about their future career orientation [58].

Cognitive skills have always been overestimated compared to other aspects, such as emotional and social ones. Research demonstrates that educational outcomes are closely related to emotional intelligence, and in-school pedagogical guidance should include good classroom management to later design emotionally based learning environments [59]. The traditional school system mainly depends on visual and verbal intelligences; however, a study of 168 students has revealed that young people prefer active learning, which involves movement and emotional experiences, such as kinesthetic and musical intelligences. A reason for the decline in academic interest may be related to disappointing results achieved in traditional types of evaluation. Children may become discouraged and lose confidence in their cognitive skills, which in turn leads them to lose their professional interest and values [60]. Closely related to school motivation is school satisfaction, which can be stimulated by a modern teaching style, the integration and optimization of educational resources in an original way and the adaptation of lessons to the needs and expectations of students [61].

The relationship between cognition and motivation has an impact on the effectiveness of learning strategies, which justifies children’s options for different school or leisure activities. The sports coach’s verbal encouragement provides a sense of well-being, optimism, and motivation for success, and this attitude is transferred to the school environment. The idea of performance implemented through sport is more effective for school success than classical methods existing in the educational system, and teachers should use the verbal encouragement method as a form of stimulating the learning process [62].

### 1.4. The Current Study

The aims of the research were to investigate the benefits of dancesport for the development of institutionalized children’s learning strategies, to examine gender-dependent differences in learning strategies, as well as to formulate possible recommendations regarding the practice of dance at the age of preadolescence, from the perspective of school success vectors

The following research questions were added: 

1. What are the benefits which dance practice brings to institutionalized children’s learning strategies?

2. Which are the gender-dependent differences in the case of institutionalized children, in terms of learning strategies?

## 2. Materials and Methods

### 2.1. Sample

The study involved 30 institutionalized children without intellectual disabilities: 17 girls and 13 boys, aged 11–12 years old (9 girls aged 11, 8 girls aged 12 years old, 4 boys aged 11, and 9 boys aged 12 years old), who did not participate in extracurricular physical activities. They were included in the target group after consulting the specialized staff from the following four foster homes in Romania, Constanta County: “Antonio”, “Micul Rotterdam” (Little Rotterdam), “Delfinul” (The Dolphin), and “Callatis”. The participants were selected based on the following criteria:-Children without physical or mental disabilities;-Students benefiting from other educational programs provided by the institutionalized system;-Children aged between 11 and 12 years old.

Children in the target group are at the age of preadolescence, a period of changes that can influence the lifestyle and professional path of young people. Stimulating learning strategies and improving school success vectors at this age can have positive effects on the quality of life of institutionalized children. The preadolescents who participated in this study were assessed according to the characteristics specific to the age of 11–12 years, with the School Motivation and Learning Strategies Inventory highlighting different scores depending on gender and the age range of the participants. 

The School Motivation and Learning Strategies Inventory was applied by the authors of the present study in collaboration with the specialized staff of the mentioned foster homes, following direct contact with the children.

### 2.2. Instruments

We selected the School Motivation and Learning Strategies Inventory (SMALSI), created by Stroud and Reynolds (2006) [63]. SMALSI is a standardized test of the PEDb computerized platform created by Cognitrom (a Romanian psychological assessment company), with the inventory being validated for the Romanian population. This platform is a software application that measures individual development and mental health, and provides career counseling; it consists of psychological tests and resources for the assessment and remediation of mental health problems, but also for the school and vocational guidance of children and adolescents.

This method of assessing school motivation is recommended by psychologists because it has a lot of strengths and can be applied both before a specialized program and at the end of the intervention period [63]. In most cases, this test is used to identify the achievement levels of students and the causes of their poor skills, but also to guide teaching methods towards correcting the problems [64]. 

SMALSI is a self-report inventory that measures nine areas associated with learning strategies, six of which focus on student strengths, and three which are aimed at student liabilities (Table 2). Additionally, an inconsistent responding index (INC) is automatically generated. The INC scale indicates to what extent the student has provided inconsistent, random responses due to inattention, carelessness, lack of understanding of the item content, or other disrupting factors. If the index score is greater than 5, the test is considered invalid. No students were removed from the research as a result of their responses to INC scale.

The response options for this inventory are never, sometimes, often, and almost always (a 4-step Likert scale), and during the processing stages, these responses generate a raw score which is transformed into a T value (eliminating age and gender differences), thus obtaining a profile of the child. SMALSI is part of the PEDb computerized platform, created by Cognitrom (a Romanian company for psychological assessment), the inventory being validated (and calibrated) for the Romanian population. SMALSI consists of 147 items (the questionnaire version up to and including 12 years old). The profile is automatically generated and the interpretation of the T scores is as follows: 100–61—a high score; 60–56—slightly above average, 55–45—an average result; 44–40—slightly below average; and 39–0 represents a low result. 

### 2.3. Procedure 

Previous studies have demonstrated the effectiveness of dance for the harmonious physical, mental, intellectual, social, and emotional development of preadolescent children. Based on the Theory of Multiple Intelligences, we applied a dancesport program intended for the institutionalized children taking part in the study, which was designed so as to be adapted to different types of intelligence. Since dancesport is a complex sport, it can simultaneously improve several aspects of development, leading to the development of the individual.

The difficulty of the steps and the variety of dance styles used in this study contribute to the bio-psycho-social development of each participant as a result of dance classes; in the context in which this sport stimulates the musical ear, requires logic and communication between dancers or with the audience, or can be a way of expression and release, it helps develop psychomotor skills, thus providing physical and mental benefits. These positive effects also influence social behavior and achievements in other school or leisure activities.

The intervention took place over a period of six months (May 2022–November 2022) with a frequency of two lessons per week, with each lesson lasting 60 min, and aimed to increase school motivation and results considering the learning strategies institutionalized children use. Through the School Motivation and Learning Strategies Inventory (SMALSI), we could observe the changes produced in children from the initial phase to their final assessment.

The main methods recommended for the dancesport program combine stimulation techniques through combined arts, such as stimulation through movement, music, and play-based visual and theatrical arts. They have the role to develop the participants’ social and emotional sides, improve their communication and concentration skills, stimulate their body expression and plasticity, and develop their rhythmicity and esthetic sense.

The dancesport lesson is in accordance with the sports training lesson and has the following structure:The preparatory part (15 min)—consists of organizing the group and selectively influencing the musculoskeletal system.The fundamental part (40 min)—contains the lesson themes specific to dances from the European (Standard) and Latin-American categories, which include five dances each: Waltz, Viennese Waltz, Tango, Slow Foxtrot, and Quickstep; and Samba, Cha-Cha-Cha, Rumba, Paso Doble, and Jive.The final part (5 min)—consists of stretching exercises and muscle relaxation techniques, accompanied by specific music (nature sounds, classical music genre).

The proposed program used dancesport to stimulate the skills underlying the development of different types of intelligences so that the final results of institutionalized children could be significantly improved compared to initial testing. We also aimed to improve the emotional state of children and increase their levels of interest in curricular activities, through imitation, improvisation, and composition, as working methods used to help children discover their qualities and pleasures. 

Principles for developing the dancesport program:The program includes several dance styles with varied rhythm and different emotional impact.The program starts with motor skills exercises, rhythmic exercises, and exercises focused on highlighting the esthetics of movement.The lessons aim to improve group communication and expression in society.The program provides art therapy exercises, which are recommended for improving the emotional state and training the skills that underlie the development of various types of intelligences.

### 2.4. Statistical Analysis 

Considering that the results were collected from a single sample of participants, the *t* test for paired samples was used. Following this analysis, it was determined whether the mean score of a measurement is statistically significantly different from the mean score obtained in another measurement. Cohen’s *d* determines the effect size or magnitude, which is interpreted as follows: around 0.20—the effect is small; 0.50—the effect is medium; and over 0.80—the effect is large. The *t* test and Cohen’s *d* show the effectiveness of the dance program for the improvement of analyzed learning strategies and the impact produced by this activity on the development of children participating in the study. The Mann–Whitney U test was used to see whether there are significant differences between boys and girls in terms of the learning strategies they use, before and after the intervention through dance practice. Add r (effect size), interpretation, and cite. Additionally, through the Kruskall–Wallis (H) test and the Dwass–Steel–Critchlow–Fligner pairwise comparison test, the differences between the four groups formed were examined—11-year-old boys vs. 12-year-old boys vs. 11-year-old girls vs. 12-year-old girls. 

## 3. Results

Table 3 shows the mean, standard deviation, and standard error for the two testing phases in terms of learning strategies: T0—the initial test and T1—the final test (the mean was presented for the entire sample, but also by gender, while SD and SE are presented for the whole sample). The results are expressed in T scores.

Table 3 shows increased means in the final test (T1), especially for the following variables: Study, Note, Read, Write, Test, and TIMORG. The biggest improvements were recorded for concentration during test-taking and academic interest, which indicates better emotional control in stressful moments and greater interest in achieving good school results. This is also confirmed by the better results obtained for the LOMOT, TANX, and CONDIF variables where the mean scores decreased, the largest difference being for concentration difficulties (T0 = 64.90 and T1 = 63.93, respectively). The results at group level are, generally, on average (between 45 and 55 T scores). For the CONDIF scale, a high value is highlighted, reflecting a reduced level of attention concentration to required tasks and a lower ability to ignore distractions.

The *t* test for paired samples is based on the assumption that a pairwise correlation should exist between the two sets of scores. The two sets of scores obtained for the same group of participants measured twice at different times are likely to correlate with each other also because some participants will consistently tend to have higher scores for that variable, while others will consistently tend to have lower scores. 

Table 4 describes the results of the *t* test for paired samples, highlighting the variables that have recorded significant differences between the two testing phases. The significance level indicates the lowest values for Study (0.010), effectiveness of test-taking strategies (0.005), and concentration difficulties (0.012), which means that there are significant differences between initial and final testing. In the case of note-taking, writing, and time management/organization techniques for test anxiety and academic motivation, the differences between the two phases are also significant (*p* < 0.05). 

The statistical analysis performed with the Cohen’s d test establishes the effect size resulting from the study, which is highlighted in Table 5. Except for the TANX variable (0.740), all variables display values above 0.80, meaning a large effect, especially for Test (3.014), Study (2.574), and LOMOT (1.610). These results demonstrate the impact of the dance program on academic motivation and effectiveness during school hours (Figure 1).

The Mann–Whitney’s U test was used to examine whether there are significant differences between boys and girls in terms of learning strategies, before and after the intervention through dance practice. 

Data in Table 6 highlight that the intervention through dance practice had a relatively equal effect on both groups—boys and girls (the differences between boys and girls remained almost the same after dance practice, taking into account the significance thresholds). Only in the case of the study strategies, girls obtained significantly better results than boys (*p* = 0.16 and r = 0.43 at initial testing, and *p* = 0.021 and r = 0.42 at final testing). In the case of the writing/research skills scale, a change—from a marginally significant difference (*p* = 0.054, r = 0.35) to a significant difference at the final test (*p* = 0.040, r = 0.37)—was observed. Girls registered better scores than boys in the case of learning strategies, before and after the experimental intervention (except for the LOMOT, TANX, and CONDIF scales), for example: *M*_Study strategies_ = 57.6 and SD = 8.93 (girls), and *M*_Study strategies_ = 45.9 and SD = 14.3 (boys)—at initial testing; *M*_Study strategies_ = 58.5 and SD = 9.69 (girls), and *M*_Study strategies_ = 47.5 and SD = 14.2 (boys)—at final testing; *M*_Writing_ = 51.5 and SD = 12.31 (girls), and *M*_Writing_ = 41.0 and SD = 13.4 (boys)—at initial testing; and *M*_Writing_ = 52.4 and SD = 12.66 (girls), and *M*_Writing_ = 41.7 and SD = 12.5 (boys)—at final testing. 

Not least, through the Kruskall–Wallis (H) test (Table 7), the differences between the four groups formed were investigated: 11-year-old boys vs. 12-year-old boys vs. 11-year-old girls vs. 12-year-old girls, before and after the experimental intervention through dance practice. The Dwass–Steel–Critchlow–Fligner (DSCF) pairwise comparison test was then used to verify the differences between the groups, analyzed two by two (Table 8). This analysis was carried out to capture even the most subtle differences between boys and girls of slightly different ages.

Data in Table 7 and Table 8 show that the intervention through dance practice had a relatively equal effect on all four groups, regardless of age and gender: 11-year-old boys, 12- year-old boys, 11-year-old girls, and 12-year-old girls (the differences between the investigated groups remained almost the same after dance practice—taking into account the significance thresholds). Girls registered better scores than boys in the case of learning strategies (except LOMOT and CONDIF scales) before and after the experimental intervention, regardless of age and gender, for example: *M*_Study strategies_ = 58 and SD = 8.85 (11-year-old girls), and *M*_Study strategies_ = 57.3 and SD = 9.62 (12-year-old girls)—at initial testing; *M*_Study strategies_ = 58.4 and SD = 9.44 (11-year-old girls), and M_Study strategies_ = 58.5 and SD = 10.6 (12-year-old girls)—at final testing; *M*_Study strategies_ = 39.5 and SD = 8.58 (11-year-old boys), and *M*_Study strategies_ = 48.8 and SD = 15.8 (12-year-old boys)—at initial testing; and *M*_Study strategies_ = 39.5 and SD = 8.58 (11-year-old boys), and M_Study strategies_ = 51.1 and SD = 15.3 (12-year-old boys)—at final testing. The least progress (after examining descriptive statistics) was recorded in the case of 11-year-old boys for the following scales: study strategies, note-taking, reading, writing, TIMORG, LOMOT, and CONDIF (however, we mention the very small number of institutionalized children aged 11). Additionally, generally (regardless of age), girls and boys registered almost the same progress when talking about learning strategies (see Table 3) after the intervention through dance practice.

## 4. Discussion

The stimulation of multiple intelligences through dancesport and their connection with learning strategies generated the research questions of the current study. More specifically, we were interested in highlighting the benefits which dance practice brings to institutionalized children’s learning strategies, creating a foundation for the improvement of their academic performance and school integration. Additionally, the present study aimed to underline gender-dependent differences in the case of institutionalized children in terms of learning strategies.

The lack of a family setting and adult attention to the education of institutionalized children is reflected in their school performance and academic interest. Preadolescents have average results, at group level, for: study strategies, note-taking, reading/comprehension strategies, writing/research skills, test-taking strategies, time management/organization techniques, academic motivation, and test anxiety, and are characterized by concentration difficulties (a higher score was observed). Taking into account student’s self-perception of skills in fulfilling academic tasks, as well as self-monitoring attention, specialists should intervene in order to increase attention concentration capacity. Additionally, institutionalized children (and not only them) could continue to systematically practice dance (as well as other systematic physical activities), because its beneficial role on children’s learning strategies been observed.

Specialized studies often associate school results with the types of intelligence [65]. A student who benefits from a differentiated teaching system based on the types of intelligence has several advantages: optimal learning time, stability of learning results, and ability to apply what they have learned in practice [66].

The SMALSI is a test recommended by specialists for the psychological assessment of students with learning problems, because it addresses school motivation, an aspect that is often neglected in educational assessments [63]. A group of 144 ninth-grade Romanian high school students was tested with the SMALSI. The results have shown that students attending humanities programs are more concerned with test-taking strategies, and pay attention to questions and keywords, while those who have an aptitude for science or mathematics apply strategies during the learning process. From the multitude of information received in school, they are interested in retaining only the important details, making connections and understanding the lesson. Math enthusiasts are more focused on time management than humanities students and develop organization techniques for school or extracurricular activities, thus becoming more efficient. These aspects have a positive influence on the academic performance of children who attend science and math programs [67]. Studies in this field report that females have better reading strategies than males, especially those who are dominant in musical and interpersonal intelligences, because they use problem-solving strategies more effectively [68]. These findings were confirmed by the present study in the case of pre-teens (girls having better scores for reading/comprehension strategies, than boys).

The training plan using dancesport as an educational resource implemented in this study contributed to improving the results of institutionalized children, with the differences between the initial test and the final test in terms of the eight learning strategies being significant. Thus, their school motivation increased, especially their academic interest and attention during test-taking, while their score for concentration difficulties decreased.

When talking about gender-dependent differences, girls obtained significantly better results than boys only in the case of study strategies and writing/research skills. Additionally, girls registered better scores than boys in the case of other learning strategies before and after the experimental intervention, except for the LOMOT, TANX, and CONDIF scales. Therefore, boys experience less anxiety symptoms caused by any evaluative method for their knowledge or performance, are more motivated to engage and succeed in various academic tasks, and consider that they have a higher level of attention concentration to required tasks and a better ability to ignore distractions (it is important to mention that these differences are not statistically significant). Statistical analysis of the data revealed that the intervention through dancesport practice had a relatively equal effect on both groups—boys and girls—as the gender-dependent differences remained almost the same after dance practice (taking into account the significance thresholds).

After applying the School Motivation and Learning Strategies Inventory, differences were found between girls and boys regarding their interest in study, academic motivation, concentration difficulties, and test anxiety [57]. Thus, girls seem to be more concerned with studying, more attentive during classes, and more motivated than boys, but they experience anxiety problems. These results are partially confirmed by the present research (and let us not forget that for the LOMOT and CONDIF scales, the differences between boys and girls were not statistically significant). Another study suggests that differences may also exist in relation to ethnicity, nationality, speaking skills, or the presence of disabilities. Research shows that learning strategies and time management are better developed in Asian than Hispanic students, but there are no significant differences between healthy children and children with disabilities; the latter may learn in mainstream schools and should be supported and encouraged [69]. The connection between these studies and our research is that, although there may be differences between children from different peoples and cultures, children from disadvantaged backgrounds should enjoy equal opportunities, because their school results can be as good as those of normally developing children from families. In this context, dance can be a means of combatting racism and providing integration in school environments.

The present study supports the idea that learning strategies can be associated with the teaching means and techniques specific to dance, and that dance can be used as a leisure educational resource because those who practice it benefit by improving their school performance. Dance is a representative artistic field due to its varied and complex nature that enhances cognitive and non-cognitive skills and facilitates the development of creativity, sociability, and originality. It creates a foundation for the development of learning strategies and therefore can improve school performance, with specialists mainly referring to memory and metacognition [70]. All these effects generate significant differences between dancers and non-dancers, which is why we recommend it as an educational resource for improving learning strategies. These effects can be complementary to the positive changes in children’s mental health well-being [39].

The variety of this sport can motivate a person to become physically active because it is more than art and more than leisure—it is an activity that should be practiced throughout life [11].

A comparison of the initial test results with those achieved in the final test highlights the effectiveness of the dance program designed for this research, confirming that dance can be a means of social integration for institutionalized children, promoting the acquisition of important behaviors, which increase the chances of integration in school and social environments due to the exploitation of multiple intelligences. The literature provides detailed information about the importance of educational programs based on the Theory of Multiple Intelligences for improving school outcomes. Many studies associate dance with stimulating the types of intelligence; however, this concept is not well known in Romania.

This present research has certain limitations, with maybe the most important being the reduced number of participants (especially for 11-year-old boys). Even if the sample is not too large to generalize results, the strengths of the current research have to do with the few studies carried out so far considering the learning strategies of institutionalized pre-teens. Further studies need to investigate, also, the impact of a dance program on the level of development, motivation, and learning strategies of children from organized families. When talking about institutionalized children, the results could be different if children of other ages were investigated, or if the research would have taken place in another country. Additionally, a self-report tool was used, and the obtained results are declarative, with the problem of possible desirable answers being known. Children’s academic performance could be examined, also, in relation to learning strategies. Not least, the conclusions might be different if experimental interventions (to increase academic motivation, develop learning strategies) were carried out by other means, specific to other sports disciplines.

## 5. Conclusions

The chances of developing intelligence types increase with the practice of dance from an early age, as it acts on cognitive reserves [30]. Such activities appear to support brain activity and reduce the rate of brain aging by stimulating attention, coordination, and creativity, but also by improving the activity of the cardiorespiratory system [71].

Previous studies demonstrate the effectiveness of dance for the harmonious development of preadolescents from a physical, mental, intellectual, social, and emotional point of view.

Based on the Theory of Multiple Intelligences, the dancesport program for the institutionalized children participating in this study is designed so as to be adapted to all types of intelligence theorized so far. Given that dancesport is a complex sport that includes physical, cognitive, and social elements [72], it has the quality of simultaneously improving several sides of development, which contributes to the overall development of an individual, influencing social behavior and the outcomes of school and leisure activities.

The conclusions of the current study underline the benefits of dancesport as an educational resource for the development of institutionalized children’s learning strategies. In the case of study strategies, note-taking, reading/comprehension strategies, writing/research skills, test-taking strategies, time management/organization techniques, academic motivation and test anxiety, the results (at group level) are, generally, within average limits, before and after the intervention through dancesport practice (higher values were observed, however, for the CONDIF scale). Significant improvements were recorded for concentration during test-taking and academic interest. Additionally, better results were obtained for the TANX and CONDIF scales, where the mean scores significantly decreased, with the largest difference being for concentration difficulties. Considering the gender-dependent differences, girls registered significantly better results than boys only in the case of study strategies and writing/research skills. Not least, the intervention through dancesport practice had a relatively equal effect on children’s learning strategies regardless of gender.

This study highlights the importance of dancesport for the development of institutionalized children`s learning strategies and various types of intelligence, and the results can represent possible benchmarks in future studies regarding inclusive education and practice of dancesport at the age of preadolescence, from the perspective of school success vectors.

## Figures and Tables

**Figure 1 children-10-01039-f001:**
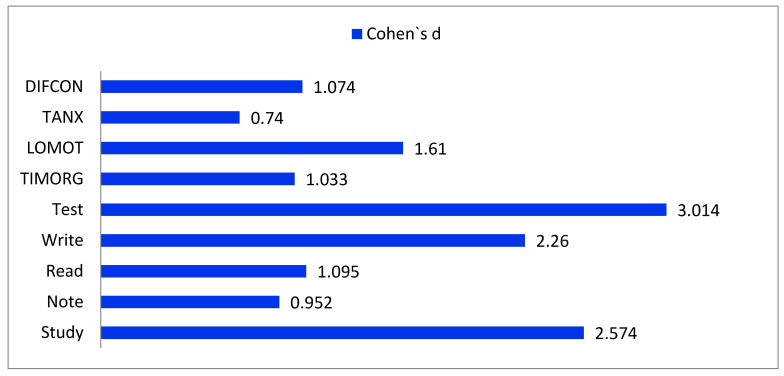
Effect of the dance program on learning strategies.

**Table 1 children-10-01039-t001:** Including multiple intelligences tasks in children’s education.

Type of Intelligence	Learning Strategies
Verbal/linguistic	Learning by listening, storytelling, and reading literary works that stimulate imagination and fiction stories that include role play and creative processes.
Logical/mathematical	Learning from stories about discoveries, research, or real processes.
Visual/spatial	Learning by drawing or reading narratives, optical illusions, visual analysis, videos, and online programs.
Bodily/kinesthetic	Learning by dancing in a variety of styles that tell a story through body movements.
Musical/rhythmic	Learning through lyrics by setting words to musical rhythm or listening to songs with educational themes.
Interpersonal/social	Learning by accessing information about personalities, studying their biographies, or finding out details of their personal and professional lives.
Intrapersonal/emotional	Learning through the journaling technique, and writing about personal feelings and experiences either on paper or as electronic journals or video recordings.
Naturalistic	Learning through fables whose characters are animals, imaginary creatures, or plants that tell stories with a powerful moral message.

Authors’ own processing.

**Table 2 children-10-01039-t002:** Description of SMALSI scales.

Variables	Description
Study strategies (STUDY)	The STUDY scale assesses the student’s ability to develop and apply learning, memory, and encoding strategies, identify important information and use a variety of resources.
Note-taking/listening skills (NOTE)	The NOTE scale assesses the student’s ability to discriminate important from irrelevant information in the classroom, organize the note-taking process, and be efficient when taking notes.
Reading/comprehension strategies (READ)	The READ scale assesses the student’s ability to develop and apply a number of reading and comprehension strategies.
Writing/research skills (WRITE)	The WRITE scale assesses the student’s ability to conduct research on various topics, access and record relevant information, and organize search results and then summarize them in written form.
Test-taking strategies (TEST)	The TEST scale assesses the student’s knowledge and ability to apply test-taking strategies, which reflects their level of preparation.
Time management/organization techniques (TIMORG)	The TIMORG scale assesses the student’s ability to manage and use time effectively as well as organize classroom study materials and homework.
Low academic motivation (LOMOT)	The LOMOT scale assesses the student’s lack of motivation to engage and succeed in various academic tasks.
Test anxiety (TANX)	The TANX scale assesses the student’s experience of anxiety symptoms caused by any evaluative method for their knowledge or performance.
Concentration/attention difficulties (CONDIF)	The CONDIF scale assesses the student’s self-perception of skills in fulfilling academic tasks, attention to required tasks, and ignoring distractions.

**Table 3 children-10-01039-t003:** Descriptive statistics of SMALSI results.

Variables		Mean	SD	SE
1	Study T1	53.73 (Mg = 58.5, Mb = 47.5)	12.92	2.35
	Study T0	52.57 (Mg = 57.6, Mb = 45.9)	12.77	2.33
2	Note T1	48.10 (Mg = 51.2, Mb = 44.1)	12.70	2.32
	Note T0	47.80 (Mg = 51.1, Mb = 43.6)	13.03	2.38
3	Read T1	51.81 (Mg = 55.5, Mb = 47.0)	11.49	2.09
	Read T0	51.62 (Mg = 55.1, Mb = 47.0)	11.45	2.09
4	Write T1	47.77 (Mg = 52.4, Mb = 41.7)	13.48	2.46
	Write T0	46.93 (Mg = 51.5, Mb = 41.0)	13.64	2.49
5	Test T1	45.30 (Mg = 46.6, Mb = 43.5)	11.61	2.12
	Test T0	43.70 (Mg = 45.4, Mb = 41.5)	11.67	2.13
6	TIMORG T1	49.81 (Mg = 53.9, Mb = 44.4)	12.75	2.32
	TIMORG T0	49.40 (Mg = 53.2, Mb = 44.4)	12.78	2.33
7	LOMOT T1	50.90 (Mg = 53.0, Mb = 48.2)	11.25	2.05
	LOMOT T0	51.50 (Mg = 53.8, Mb = 48.5)	10.922	1.99
8	TANX T1	53.37 (Mg = 53.9, Mb = 52.7)	9.386	1.71
	TANX T0	53.63 (Mg = 54.2, Mb = 52.9)	9.34	1.70
9	CONDIF T1	64.6 (Mg = 65.7, Mb = 62.0)	12.70	2.31
	CONDIF T0	64.1 (Mg = 66.1, Mb = 62.6)	12.33	2.25

Note: T1: final testing; T0: initial testing; Mg: mean girls; Mb: mean boys; SD: Standard Deviation; SE: Standard Error Mean.

**Table 4 children-10-01039-t004:** Dependent *t* test results.

Variables		Paired Differences					*t*	*p*
		MD	SD	SE	95% CI			
					Lower	Upper		
1	Study T1–Study T0	−1.167	2.574	0.470	−2.128	−0.205	−2.482	0.010 *
2	Note T1–Note T0	−0.300	0.952	0.174	−0.656	0.056	−1.725	0.048 *
3	Read T1–Read T0	−0.200	1.095	0.200	−0.609	0.209	−1.000	0.163
4	Write T1–Write T0	−0.833	2.260	0.413	−1.677	0.011	−2.019	0.026 *
5	Test T1–Test T0	−1.500	3.014	0.550	−2.626	−0.374	−2.726	0.005 **
6	TIMORG T1–TIMORG T0	−0.367	1.033	0.189	−0.753	0.019	−1.943	0.031 *
7	LOMOT T1–LOMOT T0	0.600	1.610	0.294	−0.001	1.201	2.041	0.025 *
8	TANX T1–TANX T0	0.267	0.740	0.135	−0.010	0.543	10.975	0.029 *
9	CONDIF T1–CONDIF T0	0.467	1.074	0.196	0.066	0.868	2.379	0.012 *

Note: * *p* < 0.05, ** *p* < 0.01, MD: Mean Difference; *p*: significance level.

**Table 5 children-10-01039-t005:** Effect of the dance program on learning strategies at institutionalized children.

Variable			Standardizer	Point Estimate	95% Confidence Interval	
					Lower	Upper
1	Study T0–T1	Cohen’s d	2.574	−0.453	−0.826	−0.073
		Hedges’ correction	2.643	−0.441	−0.804	−0.072
2	Note T0–T1	Cohen’s d	0.952	−0.315	−0.679	0.054
		Hedges’ correction	0.978	−0.307	−0.662	0.053
3	Read T0–T1	Cohen’s d	1.095	−0.183	−0.542	0.180
		Hedges’ correction	1.125	−0.178	−0.528	0.175
4	Write T0–T1	Cohen’s d	2.260	−0.369	−0.736	0.004
		Hedges’ correction	2.321	−0.359	−0.717	0.004
5	Test T0–T1	Cohen’s d	3.014	−0.498	−0.874	−0.114
		Hedges’ correction	3.095	−0.485	−0.851	−0.111
6	TIMORG T0–T1	Cohen’s d	1.033	−0.355	−0.721	0.017
		Hedges’ correction	1.061	−0.346	−0.702	0.017
7	LOMOT T0–T1	Cohen’s d	1.610	0.373	−0.001	0.740
		Hedges’ correction	1.654	0.363	−0.001	0.721
8	TANX T0–T1	Cohen’s d	0.740	0.361	−0.012	0.727
		Hedges’ correction	0.760	0.351	−0.012	0.708
9	CONDIF T0–T1	Cohen’s d	1.074	0.434	0.056	0.806
		Hedges’ correction	1.103	0.423	0.055	0.785

**Table 6 children-10-01039-t006:** Learning strategies—differences between boys and girls.

SMALSI	Boys vs. Girls—Before and after the Intervention through Dance Practice							
	Before	After	Before	After	Before	After	Before	After
	U		*p*		z		r	
Study	52.7	55	0.016	0.021	2.41	2.30	0.43	0.42
Note	73	72.5	0.121	0.116	1.54	1.57	0.28	0.29
Read	74	70	0.131	0.094	1.50	1.67	0.27	0.30
Write	64	61	0.054	0.040	1.92	2.05	0.35	0.37
Test	94.5	100.5	0.516	0.690	0.65	0.40	0.12	0.07
TIMORG	71.5	70	0.107	0.094	1.61	1.67	0.29	0.30
LOMOT	83.5	86	0.267	0.315	1.11	1.00	0.20	0.18
TANX	103	101	0.769	0.706	0.29	0.37	0.05	0.07
CONDIF	94	92.5	0.502	0.463	0.67	0.73	0.12	0.13

Note: U: test Mann-Whitney values; *p*: threshold; z: z-score is automatically generated when running U test procedure, taking into account the U values and the sample size (z-score was used in calculating the effect size); r: effect size

**Table 7 children-10-01039-t007:** Learning strategies according to age and gender.

SMALSI	Institutionalized Children’s Results According to Age and Gender—Before and after the Intervention through Dance Practice					
	χ^2^		*p*		ε^2^	
	Before	After	Before	After	Before	After
Study	7.84	7.55	0.050	0.056	0.270	0.260
Note	5.16	5.71	0.161	0.127	0.177	0.196
Read	3.58	4.12	0.311	0.249	0.123	0.142
Write	4.57	5.32	0.206	0.150	0.157	0.183
Test	2.75	2.03	0.431	0.565	0.094	0.070
TIMORG	3.24	3.56	0.357	0.313	0.111	0.122
LOMOT	2.38	2.19	0.497	0.534	0.082	0.075
TANX	1.11	1.44	0.774	0.697	0.038	0.049
CONDIF	1.51	1.72	0.680	0.631	0.052	0.059

Note: χ^2^: Kruskall–Wallis (H) test value; *p*: threshold; ε^2^: effect size.

**Table 8 children-10-01039-t008:** DSCF test results for pairwise comparisons—learning strategies according to age and gender (only the closest thresholds to *p* = 0.05 are presented).

Groups Age and Gender		Before the Intervention through Dance Practice	
		W	*p*
		Study	
1	3	−3.501	0.064
2	3	−3.128	0.120
		Note	
1	3	−3.068	0.132
		After the Intervention through Dance Practice	
		W	*p*
		Study	
1	3	−3.496	0.064
2	3	−3.133	0.119
		Note	
1	3	−3.068	0.132

Note: 1–11-year-old girls; 2–12-year-old girls; 3–11-year-old boys; 4–12-year-old boys; W—Wilcoxon value.

## Data Availability

Data available on request due to restrictions eg privacy or ethical. The data presented in this study are available on request from the corresponding author. The data are not publicly available due to the study was carried out with the consent of General Directorate of Social Assistance and Child Protection, and data about institutionalized children should be confidential.

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
