# Peer review of "Using Dancesport as an Educational Resource for Improving Institutionalized Children’s Learning Strategies"

_children, 2023, doi:10.3390/children10061039_

Round 1

Reviewer 1 Report

see attachment.

Author Response

The concept of dance in the article is very vague and does not specify which dance project it is. Significant stylistic differences exist between dance projects, and the training methods are also very different. The effects presented after practice will also be different. For example, sports dance, which leans towards competitive dance and combines two elements: "sports competition" and "dance". For example, ballet tends to be more ornamental and lacks the concept of sports competition. The discussion provides a detailed description of the effects of dance, but there is less discussion of the reasons for how dance actually produces these effects.

                Although we understood the impact of dance practice on the differences in learning ability between boys and girls, we did not understand why the authors separated the age groups of 11 and 12 years. Some of the information from the comparison of boys and girls in the 11 and 12-year-old groups also does not seem to be reflected in the conclusions.

Line14-17             The study context does not provide enough clarity about the background or significance of the study to know why the study is important. The research question or hypothesis of the study is not clearly stated, which makes it difficult for the reader to understand the specific purpose of the study.

Specialized studies mention that extracurricular activities (including dance) contribute to the stimulation of multiple intelligences on whose development the educational process and academic success depend. The aims of the study were to investigate the benefits of dance for the development of institutionalized children’s learning strategies and to examine gender-dependent differences, as well as to formulate possible recommendations regarding the practice of dance at the age of preadolescence, from the perspective of school success vectors.

Line18-22             The methodology of the study is not clear, and it is not known how the intervention was carried out to understand how the study was conducted. No information was mentioned about the adequacy of the sample size, the reliability and validity of the study, and it was not known whether the evaluation indicators were reasonable to assess the quality and credibility of the study.

Through the School Motivation and Learning Strategies Inventory (SMALSI), we could observe the changes produced in children from the initial phase to the their final assessment at the end of a dance program. The intervention took place over a period of six months with a frequency of two lessons per week, with each lesson lasting 60 minutes, and aimed to increase school motivation and performance, considering the learning strategies used by institutionalized children.

Thirty institutionalized children, aged 11-12 years, participated in the research, during which they did not engage in other extracurricular physical activities. The preadolescents were assessed using the School Motivation and Learning Strategies Inventory (SMALSI).

Line23-28              The results do not correspond to the discussion, there is no conclusion, and the author should have used the wrong words in the discussion, and there is no guiding suggestion to make.

                Conclusion: At the end of the dance program, significant improvements in academic interest were observed due to the testing strategies used. Better results were also obtained for the scales of anxiety and difficulty concentrating during tests, where average scores decreased significantly.

Introduction

                The logic of the whole introduction part of the thesis is rather confusing and I think the following questions should be answered clearly:

                1 Why were the institutionalized children chosen and what are their characteristics?

Methods: The study involved 30 institutionalized children, without intellectual disabilities: 17 girls and 13 boys, aged 11-12 years (9 girls aged 11, 8 girls aged 12, 4 boys aged 11, and 9 boys aged 12), who did not participate in extracurricular physical activities. They were included in the target group after consulting the specialized staff from the following four foster homes in Romania, Constanta County: “Antonio”, “Micul Rotterdam” (Little Rotterdam), “Delfinul” (The Dolphin) and “Callatis”. The participants were selected based on the following criteria:

  • children without physical or mental disabilities;
  • students benefiting from other educational programs provided by the institutionalized system;
  • children aged between 11 and 12 years.

                2 Why do you think dance can improve learning strategies, and what are the specific arguments? Discussing the possibility of differences in the impact of dance on the learning strategies of males and females

                3 What are the criteria for evaluating learning strategies? How can the impact of dance on learning strategies be quantified?

Learning strategies were assessed using a standardized test from the PEDb platform. This is a software application that measures individual development and mental health, and provides career counseling; it consists of psychological tests and resources for the assessment and remediation of mental health problems, but also for the school and vocational guidance of children and adolescents.

In 1.2, the author introduces dance as an educational resource and conducts a background review. Here, the author vaguely introduces the role of dance, and dance as a form of performance that shows dancers' inner activities through expression interpretation and body movements, There are many types of dances, and different types of dances have different action forms, difficulty levels, expression methods, dance styles, emotional performances, etc. Therefore, it is recommended to elaborate on the different mechanisms of different dances. The dance intervention in this study should also explain.

In terms of visual/spatial intelligence, the individual acquires the ability to think in images, to either clearly visualize the examples provided by the instructor or abstractly visualize them based on imagination, which will be adapted to the workspace or the movement possibilities

The earlier dance education begins, the greater the chances of developing intelligence. Brain activity is extremely important in children, and this sport has a positive mental, emotional, and social impact from the first years of life, the forms of communication in that period relying on expression through gestures and movements.

Previous studies demonstrate the effectiveness of dancesport for the harmonious development of preadolescents from a physical, mental, intellectual, social, and emotional point of view. Based on the Theory of Multiple Intelligences, the dance program for the institutionalized children participating in this study is designed so as to be adapted to all types of intelligence theorized so far. Given that dance is a complex sport that includes a variety of styles from the Latin-American and European categories, it has the quality of simultaneously improving several sides of development, which contributes to the overall development of an individual.

The School Motivation and Learning Strategies Inventory used in this research highlights the changes in institutionalized children in terms of school success, contributing to the assessment of certain types of intelligence, especially linguistic, logical, and emotional intelligences.

Line30   The title of the article is "Using Dance as an Educational Resource for Improving Children's Learning Strategies". However, in the introduction section, line32-41 mentions that sports activities play an important role in children's growth and development, and dance is not entirely equivalent to sports.

Dance is a physical activity that includes various forms of expression. Its complexity and aerobic nature provide health benefits, reducing excess weight (which increases the risk of chronic illnesses such as cancer, diabetes, and cardiovascular disease), correcting posture, and strengthening muscles .

According to the literature, dancesport is, due to the countless functions it fulfills, at the border between sport and art, being a motor activity expressed nonverbally both in the sphere of bodily expression and from a narrative point of view by performing rhythmic movements based on choreographic scenarios that involve multiple movements of the body such as travels, twists, leaps, turns, flexion and extension, etc.

The variety of functions and benefits offered by sports dance organizes it into three distinct categories: elite dance, dance for all (whose purpose is to achieve an optimal state of health and harmonious physical development, educate body esthetics, develop conditional and coordination skills), and adapted dance (the compensatory effects of dance contribute to the correction of defects).

Line39-40             Please delete this section on the role of parents in education and focus here on the impact of leisure physical activities on children's health. The influence of parents should be addressed in the context of institutional child characteristics(1.1. The effects of institutionalization on children’s development).

Line85-90             Just saying the status quo is not supported by a valid theory that leads to the hypothesis and purpose of the research one wants to do.

Line114-210         With regard to dance as an educational resource for improving children's learning strategies, please provide a clear discussion of the impact of dance on learning strategies, either from the Multiple intelligences theory or in the order of the Learning Strategies Questionnaire.

Dance can be practiced with a therapeutic purpose: it improves the health status of people with various mental or motor conditions, increases school or professional results, and prevents the deterioration of brain activity. The enjoyment of music and dance plays an important role in increasing the individual’s daily motivation.

Line212-245         Please discuss the impact of dance on learning strategies from the perspective of multiple intelligences, rather than just multiple intelligences and learning strategies

Materials and Methods

Line 286 The authors mention that gender differences in learning strategies will be studied, so are there gender differences in learning strategies before the study begins? I think a relevant literature review should be included in the introduction.

Line 292-301        The determination of the sample size and the selection of the sample were not clear to see without a control group, and this method is not accurate enough to prove the title of the study.

Line 294                Why are males and females again divided into two age groups, please include a description in the introduction

Children in the target group are at the age of preadolescence, a period of changes that can influence the lifestyle and professional path of young people. Stimulating learning strategies and improving school success vectors at this age can have positive effects on the quality of life of institutionalized children. The preadolescents who participated in this study were assessed according to the characteristics specific to the age of 11-12 years, the School Motivation and Learning Strategies Inventory highlighting different scores depending on gender and the age range of the participants.

Line 311                There was no mention of the SMALSI’s reliability.

SMALSI is a standardized test of the PEDb computerized platform created by Cognitrom (a Romanian psychological assessment company), the inventory being validated for the Romanian population. This platform is a software application that measures individual development and mental health, and provides career counseling; it consists of psychological tests and resources for the assessment and remediation of mental health problems, but also for the school and vocational guidance of children and adolescents.

Line 347                Dance training program should add exercise intensity detection.

The dance lesson is in accordance with the sports training lesson and has the following structure:

  • The preparatory part (15 minutes) – consists in organizing the group and selectively influencing the musculoskeletal system;
  • The fundamental part (40 minutes) – contains the lesson themes specific to dances from the European (Standard) and Latin-American categories, which include five dances each: Waltz, Viennese Waltz, Tango, Slow Foxtrot, and Quickstep; Samba, Cha-Cha-Cha, Rumba, Paso Doble, and Jive;
  • The final part (5 minutes) – consists of stretching exercises and muscle relaxation techniques, accompanied by specific music (nature sounds, classical music genre).

Line 364                There are many styles of dance, this sentence is not clear and rewrite it what kind of dance is it.

Dancesport

Line 374                What are the independent variables in Statistical analysis? I think the independent variables seem to be gender and age group and parameters before and after the experiment, so we can also consider the method of analysis of variance。

Line 387                How are the four groups compared? The statistical methods are not clearly stated and the variables are not clearly articulated.

Results

Line 410                The analysis of the results was insufficient.

Line 420-426       The ability to read is not mentioned and can be more fully summarized and elaborated.

Discussion        

                The whole discussion did not establish a good causal relationship between dance and learning strategies; it was just talking about appearances without depth.

Line 568-572        How do findings such as the racial differences described relate to this study?

The connection between these studies and our research is that, although there may be differences between children from different peoples and cultures, children from disadvantaged backgrounds enjoy equal opportunities, and their school results can be as good as those of normally developing children from families. In this context, dance can be a means of combatting racism and integration in school environments.

Line 573-583        This study only measured the correlation between dance practice and learning strategies, and there may be other factors contributing to the observed effects.

Conclusion     

Line 610-611        It is difficult to determine if it is the dance that brings the benefits and only has a short-term effect, not knowing if it is beneficial for future development.

Reviewer 2 Report

The authors presented an article devoted to an actual and practically significant problem: the use of dances for the development of institutionalized children’s learning strategies.

The introduction contains a detailed review of the literature and justification of the research problem.

However, in my opinion, the design of the authors’ research and the presentation of its results in the article are flawed.

The most serious drawback of the study is that there is no control group in the experiment. Changes in the experimental group after intervention are small in absolute numbers and may be due to the effect of repeated testing, natural development, but not dance practice. To confirm the impact of dance practice on learning strategies, a comparison with an equivalent group of children who did not participate in dance practice is necessary.

Less serious shortcomings are related to the presentation of the study in the article:

1)    It is desirable to describe in more detail the dance program used;

2)      It is desirable to present tables in a generally accepted form, see for example the APA manual;

3)      In my opinion, due to the small number, it makes no sense to compare 4 subgroups: 11-year-old girls, 11-year-old boys, 12-year-old boys, 12-year-old girls

4)      If the article is accepted for publication, I recommend that the authors also pay attention to the title (authors need to clarify that they are talking about institutionalized children from Romania) and annotation (add a brief description of the dance practice)

Author Response

The authors presented an article devoted to an actual and practically significant problem:

the use of dances for the development of institutionalized children’s learning strategies.

The introduction contains a detailed review of the literature and justification of the research

problem.

However, in my opinion, the design of the authors’ research and the presentation of its

results in the article are flawed.

The most serious drawback of the study is that there is no control group in the experiment.

Changes in the experimental group after intervention are small in absolute numbers and

may be due to the effect of repeated testing, natural development, but not dance practice.

To confirm the impact of dance practice on learning strategies, a comparison with an

equivalent group of children who did not participate in dance practice is necessary.

We also tested a group of dancers who come from families (control group), and their results, in terms of learning strategies, were significantly better. Considering the social and educational environment from which the institutionalized children come, we considered that these differences regarding the vectors of school success are normal. The emotional state and interests of children who come from families are different compared to those who grew up in centers, without parental love and financial possibilities. For this reason, we applied the dance program to institutionalized children, highlighting the changes that occurred in terms of school motivation. Through dancesport activity, we wanted to stimulate all types of multiple intelligences, to improve emotional state and self-esteem, and to increase the interest in school and extracurricular activities. In a future study, we also want to highlight the differences between children from centers and dancing children from families.

Less serious shortcomings are related to the presentation of the study in the article:

1) It is desirable to describe in more detail the dance program used;

The dance style used in the study is dancesport, which provides two categories of dance: European and Latin American (according to the World DanceSport Federation - WDSF)

The dance lesson is in accordance with the sports training lesson and has the following structure:

  • The preparatory part (15 minutes) – consists in organizing the group and selectively influencing the musculoskeletal system;
  • The fundamental part (40 minutes) – contains the lesson themes specific to dances from the European (Standard) and Latin-American categories, which include five dances each: Waltz, Viennese Waltz, Tango, Slow Foxtrot, and Quickstep; Samba, Cha-Cha-Cha, Rumba, Paso Doble, and Jive;
  • The final part (5 minutes) – consists of stretching exercises and muscle relaxation techniques, accompanied by specific music (nature sounds, classical music genre).

2) It is desirable to present tables in a generally accepted form, see for example the APA

manual;

3) In my opinion, due to the small number, it makes no sense to compare 4 subgroups: 11-

year-old girls, 11-year-old boys, 12-year-old boys, 12-year-old girls

The scales in the questionnaire correspond to a raw score. The SMALSI test has a different score according to gender and age, therefore we considered it necessary to carry out an analysis for each category.

4) If the article is accepted for publication, I recommend that the authors also pay

attention to the title (authors need to clarify that they are talking about institutionalized

children from Romania) and annotation (add a brief description of the dance practice).

TITLE: Using Dancesport as an Educational Resource for Improving Institutionalized Children’s Learning Strategies

Reviewer 3 Report

The manuscript entitled "Using Dance as an Educational Resource for Improving Children’s Learning Strategies" is a well-structured manuscript it has an important topic. The introduction is extremely detailed. The Authors are showing the theoretical background of their study. However, I would recommend adding more studies about the relationships between learning strategies and physical activity. In the Materials and Methods sections, the readers could see a detailed view of how the study was conducted. I believe no other information needs to add to this part. The results part needs some revision. Is not clear what T1 and T0 mean please add this to clarify this in the text. I also recommend adding Cohen’s due to Table 4. Table 7 is hard to understand.  The discussion is well written the Authors conclude relevant things, but more studies on this topic could be added to strengthen the results. Overall, it is a great study!

Other suggestion: 

Please add a reference (line 65): A final meta-analysis of more than one hundred studies compared students who received 66 SEL with those who did not.

The English are fine! Proofreadin is always needed. 

Author Response

The manuscript entitled "Using Dance as an Educational Resource for Improving Children’s Learning Strategies" is a well-structured manuscript it has an important topic. The introduction is extremely detailed. The Authors are showing the theoretical background of their study. However, I would recommend adding more studies about the relationships between learning strategies and physical activity. In the Materials and Methods sections, the readers could see a detailed view of how the study was conducted. I believe no other information needs to add to this part. The results part needs some revision. Is not clear what T1 and T0 mean please add this to clarify this in the text. I also recommend adding Cohen’s due to Table 4. Table 7 is hard to understand.  The discussion is well written the Authors conclude relevant things, but more studies on this topic could be added to strengthen the results. Overall, it is a great study!

Motivation and school success depend on certain internal dynamic factors such as skills and motives to satisfy the need for success, but the training process decisively benefits from motor activities and the support provided by educators in the direction of perception, orientation, and awareness.

Through their specific content, sports activities influence the path to general education in motor, intellectual, esthetic, and emotional terms, contributing to the formation of interests and motivation for obtaining good results in other disciplines and fields, but also to the bio-psychological development of the child.

Compared to people who practice other physical and leisure activities, dancers and musicians have a greater ability to distinguish sounds, understand information, anticipate and imitate the next movements of people around them or other living organisms, feel the rhythm and synchronize movements to music, orient themselves in space and time, and control their posture. Also, they are more concerned with professional recognition and relationships with those around them.

Table 3 shows the mean, standard deviation and standard error for the two testing phases in terms of learning strategies, T0 – initial test, and T1 – final test (the mean was presented for the entire sample, but also by gender, while SD and SE are presented for the whole sample). The results are expressed in T scores.

Table 5. 

Variables

Standardizera

Point Estimate

95% Confidence Interval

Lower

Upper

1

Study T0 - T1

Cohen's d

2.574

-.453

-.826

-.073

Hedges' correction

2.643

-.441

-.804

-.072

2

Note T0 - T1

Cohen's d

.952

-.315

-.679

.054

Hedges' correction

.978

-.307

-.662

.053

3

Read T0 - T1

Cohen's d

1.095

-.183

-.542

.180

Hedges' correction

1.125

-.178

-.528

.175

4

Write T0 - T1

Cohen's d

2.260

-.369

-.736

.004

Hedges' correction

2.321

-.359

-.717

.004

5

Test T0 - T1

Cohen's d

3.014

-.498

-.874

-.114

Hedges' correction

3.095

-.485

-.851

-.111

6

TIMORG T0 - T1

Cohen's d

1.033

-.355

-.721

.017

Hedges' correction

1.061

-.346

-.702

.017

7

LOMOT T0 - T1

Cohen's d

1.610

.373

-.001

.740

Hedges' correction

1.654

.363

-.001

.721

8

TANX T0 - T1

Cohen's d

.740

.361

-.012

.727

Hedges' correction

.760

.351

-.012

.708

9

CONDIF T0 - T1

Cohen's d

1.074

.434

.056

.806

Hedges' correction

1.103

.423

.055

.785

The statistical analysis performed with the Cohen’s d test establishes the effect size resulting from the study, which is highlighted in Table 5. Except for the TANX variable (0.740), all variables display values above 0.80, meaning a large effect, especially for Test (3.014), Study (2.574) and LOMOT (1.610). These results demonstrate the impact of the dance program on academic motivation and effectiveness during school hours.

Conclusion

The chances of developing intelligence types increase with the practice of dance from an early age, as it acts on cognitive reserve. Such activities appear to support brain activity and reduce the rate of brain aging by stimulating attention, coordination, and creativity, but also by improving the activity of the cardiorespiratory system.

Previous studies demonstrate the effectiveness of dance for the harmonious development of preadolescents from a physical, mental, intellectual, social, and emotional point of view.

Based on the Theory of Multiple Intelligences, the dance program for the institutionalized children participating in this study is designed so as to be adapted to all types of intelligence theorized so far. Given that dance is a complex sport that includes physical, cognitive, and social elements, it has the quality of simultaneously improving several sides of development, which contributes to the overall development of an individual, influencing social behavior and the outcomes of school and leisure activities.

Other suggestion: 

Please add a reference (line 65): A final meta-analysis of more than one hundred studies compared students who received 66 SEL with those who did not.

This project provides a final meta-analysis of more than one hundred studies compared students who received SEL with those who did not.